# Quantum billiards with correlated electrons confined in triangular transition metal dichalcogenide monolayer nanostructures

Jan Ravnik[1,2✉], Yevhenii Vaskivskyi[1], Jaka Vodeb[1,3], Polona Aupič[1], Igor Vaskivskyi [1,4], Denis Golež[3,5,6], Yaroslav Gerasimenko [4], Viktor Kabanov[1] & Dragan Mihailovic [1,4✉]

Forcing systems through fast non-equilibrium phase transitions offers the opportunity to study new states of quantum matter that self-assemble in their wake. Here we study the quantum interference effects of correlated electrons confined in monolayer quantum nanostructures, created by femtosecond laser-induced quench through a first-order polytype structural transition in a layered transition-metal dichalcogenide material. Scanning tunnelling microscopy of the electrons confined within equilateral triangles, whose dimensions are a few crystal unit cells on the side, reveals that the trajectories are strongly modified from free-electron states both by electronic correlations and confinement. Comparison of experiments with theoretical predictions of strongly correlated electron behaviour reveals that the confining geometry destabilizes the Wigner/Mott crystal ground state, resulting in mixed itinerant and correlation-localized states intertwined on a length scale of 1 nm. The work opens the path toward understanding the quantum transport of electrons confined in atomic-scale monolayer structures based on correlated-electron-materials.

[1] Dept. of Complex Matter, Jožef Stefan Institute, Ljubljana, Slovenia. [2] Laboratory for Micro and Nanotechnology, Paul Scherrer Institut, Villigen, PSI, Switzerland. [3] Dept. of Physics, Faculty of Mathematics and Physics, University of Ljubljana, Ljubljana, Slovenia. [4] CENN Nanocenter, Ljubljana, Slovenia. [5] Dept. of Theoretical Physics, Jožef Stefan Institute, Ljubljana, Slovenia. [6] Flatiron Institute, New York, NY, USA. ✉email: jan.ravnik@psi.ch; dragan.mihailovic@ijs.si

A laser-induced quench through a first-order structural transition can create small domain structures with atomically precise shapes that are beyond the reach of current nanofabrication technologies. Viewed from a quantum physics perspective, such structures represent a fruitful playground for investigating the quantum behavior of particles in geometrically confined systems. Regular shapes, and equilateral triangles (ETs) in particular, are of special interest because they allow the study of the crossover from periodic limit cycles to chaotic trajectories[1–3]. Quantum scars—quantum interference (QI) patterns that follow traces of the paths of classical particles[4–7]—were investigated until now in fabricated mesoscopic semiconductor heterostructures and graphene by scanning gate microscopy[8,9]. However, with strongly interacting electrons such patterns are not expected due to their tendency for localization, and more elaborate structures are expected that require new theoretical approaches beyond the noninteracting electron approach. Recently, oscillating patterns were observed in interacting Rydberg atom arrays and theoretically motivated by the existence of many-body scars—manifold of low entangled excited states[10,11]. Correlation effects may be expected to give rise to perturbed trajectories in which the entanglement dynamics show a dependence on the nature of the perturbation. The observation of QI in confined correlated electron systems would be of both fundamental interest, and also of practical importance for designing coherent electron devices with correlated materials.

Here we use scanning tunneling microscopy to investigate QI in ET-shaped monolayer nanostructures of $TaS_2$ as small as ~2.6

to ~12.5 nm wide (8–38 unit cells) (Fig. 1). $TaS_2$ is a prototypical electronically correlated quasi-2D material, which is prone to carrier localization and the formation of different charge orders at different temperatures that become commensurate at 'magic' filling fractions[12].

The orthorhombic 1T polytype of $1T-TaS_2$ below ~180 K, has a commensurate charge-density-wave (CCDW) with a large modulation amplitude of $\sim 1$ electron per 13 Ta sites, localized in a $\sqrt{13} \times \sqrt{13}$ superlattice structure, which can exhibit either left (L) or right (R) chirality with respect to the crystal lattice[13] (Fig. 1a). The lattice surrounding each 13th Ta atom is distorted by the extra charge[14] resulting in the formation of a polaron. Due to Coulomb interactions between such polarons, the system is correlated and thought to be susceptible to the formation of a Mott state[14,15], whence it is often discussed in terms of a polaronic Wigner crystal[12,16,17]. The 2H (trigonal) polytype is metallic above 75 K, but forms a commensurate CCDW below this temperature, which—in contrast to the 1 T polytype—is metallic down to 1 K, below which it becomes superconducting[18].

The ET nanostructures are created by a laser pulse-induced quench through an inversion-symmetry-breaking polytype transformation of the surface atomic monolayer of a 1T polytype $TaS_2$ single crystal. The resulting ET domains are embedded laterally by a $1H-TaS_2$ crystalline layer (the 1H signifies it is a monolayer). The entire structure is epitaxial on a $1T-TaS_2$ single crystal (Fig. 1a, b)).

Understanding the behavior of correlated electrons confined in such small ETs theoretically represents a substantial challenge.

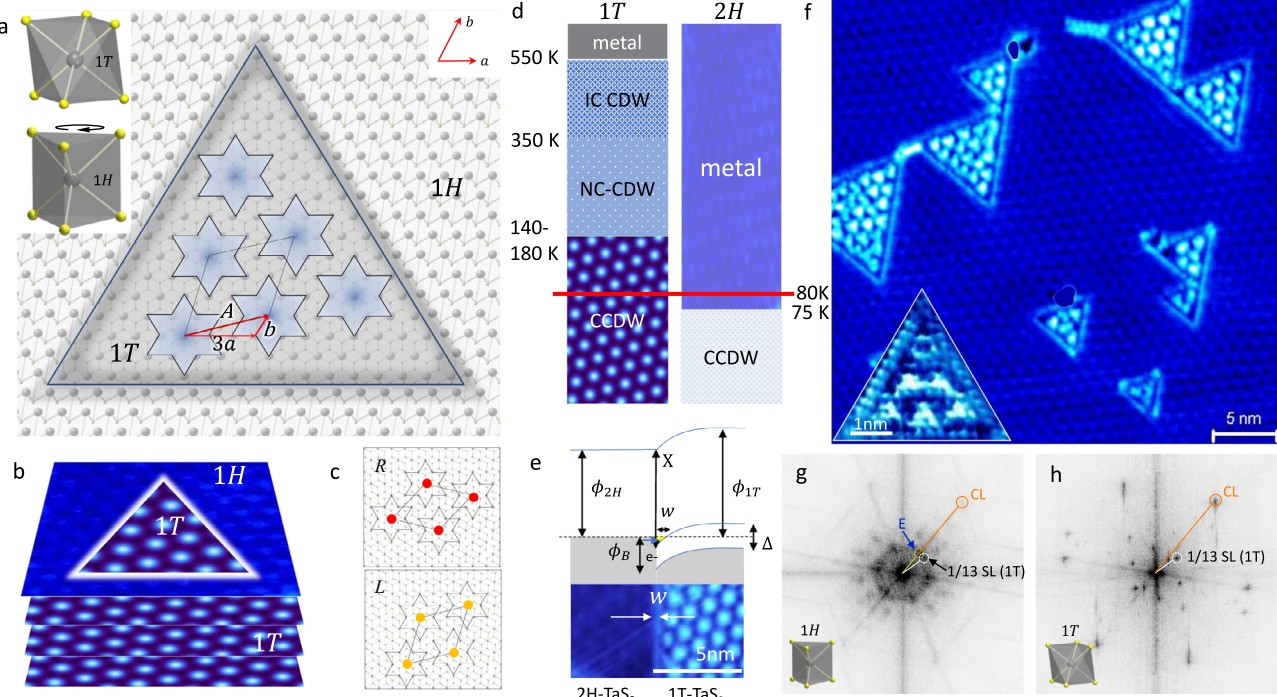

**Fig. 1 Monolayer 1T-TaS₂ structure in the shape of an ET bounded by a 1H-TaS₂ monolayer. a** A schematic picture of the left-handed packing of polarons in a C superlattice within an ET of outer dimensions 16a (inner dimensions l = 11a), where a = 0.33 Å is the lattice constant. The CCDW superlattice vector **A** is shown in terms of the lattice vectors **a** and **b**. The unit cells of the 1T and 1H crystal structures are shown in the insert. **b** The schematic structure of ETs on top of a 1T-TaS₂ substrate. **c** Ideal polaron packing in L and R-handed CCDW superlattices. **d** A phase diagram showing the charge density wave ordering transitions of the 1T- and 2H- polytypes as a function of temperature. The measurement temperature and phase transition temperatures are indicated. **e** A band diagram of the 1T-1H boundary. An edge state of width w forms within the 1T phase as a result of band bending (see Supplementary information). **f** An STM image of different sized a 1T-TaS₂ *ETs* embedded laterally within a 1H-TaS₂ monolayer on the surface of a 1T-TaS₂ single crystal. Note the ubiquitous presence of the edge state. The insert (bottom-left) shows a high-resolution image of a small ET with 6 polarons. More images are shown in the Supplementary information. **g**, **h** Fourier transforms of the 1H and 1T regions respectively. The $\sqrt{13} \times \sqrt{13}$ CCDW SL and CL peaks are indicated. A peak attributed to the polarons ordered parallel to the edge of the ETs is also indicated (E).

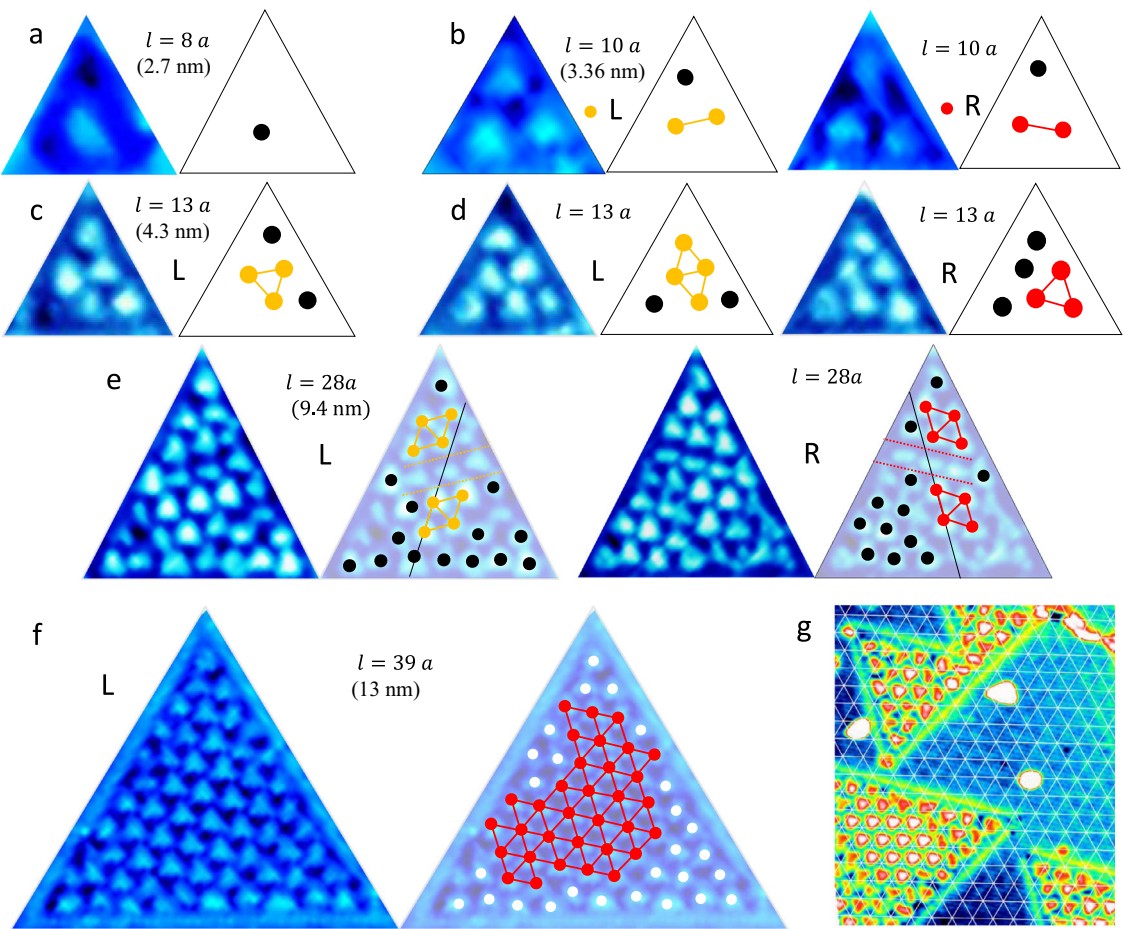

**Fig. 2 STM images of different sized ETs with l = 8...38a at 80 K with a bias voltage of 0.8 V.** Each STM image is accompanied by a schematic figure indicating the appearance of L or R chirality of the C order (yellow and red respectively). Multiple localized QI features do not conform to the C order (black dots). **a** A single dot is observed for l = 8a. **b** For l = 10a, L and R chiralities of C order appear. **c** l = 13a with 5 polarons, **d** l = 13a with 6 polarons of L and R chiralities. **e** l = 28a with L and R chiralities created nearby with the same laser pulse exposure. Similar domain wall patterns (dashed lines) are observed in both cases, as schematically shown. **f** An ET with l = 38a showing a single C domain at the center, surrounded by QPI at the edges (white dots). **g** The registry of the CCDW orders in the ETs and the 1T-TaS$_2$ layer below. The grid pattern is centered on the CCDW in the 1T layer below the 1H layer. All ETs show phase shifts of the polaron order with respect to the substrate, and to each other.

Considering CDWs as standing wave interferences formed by counter-propagating Fermi electrons[19–21] suggests an investigation using the quantum billiards (QB) approach with Fermi electrons. On the other hand, the 2D polaronic Wigner crystal picture, where the polarons are subject to Mott localization[14] suggests a correlated electron picture. Here we compare conventional QB calculations, a (classical) charged lattice gas strongly correlated electron model, and a fully quantum many-body correlated electron model using exact diagonalization methods with the aim to understand the rich variety of QI textures in both classical and quantum regimes observed by STM.

## Results

The ET structures (Fig. 1f) are created by a controlled exposure of a freshly exfoliated 1T-TaS$_2$ single crystal to laser pulses in ultrahigh vacuum at 80 K[22], where the majority of the top surface is transformed to the 1H polytype, but ET structures of 1T polytype remain structurally unchanged[23,24]. The domains have atomically defined sides parallel to the crystal axes of the 1T layer, matching the lattice structure of the surrounding 1H layer, forming a perfect ET shape with edges at 60° to each other (Fig. 1a, b). The work functions for the 1T-TaS$_2$ and 2H-TaS$_2$ polytypes are $\phi = 5.6$ and 5.2 eV respectively, so the H phase acts as a barrier for electrons in the 1T ET nanostructure[25]. At 80 K,

where the measurements are preformed, the 1H polytype is metallic, while the 1T polytype is nominally in the insulating CCDW phase[14]. (For reference, low-temperature scans showing the appearance of the $3 \times 3$ CDW of the 1T layer are shown in the Supplementary information). The H-polytype layer thus defines a confining potential barrier $\phi_B$ to the electrons inside the ET (Fig. 1e). As a result of self-organization of charges at the $1T - 1H$ interface, an edge state is formed which is clearly visible in ETs of all sizes (Fig. 1f). An interfacial band diagram based on a conventional metal-semiconductor junction is shown in Fig. 1e (see Supplementary information for details). The width $w$ of the edge state (ES) is approximately equal to the screening length in 1T-TaS$_2$, $\zeta = 1 \sim 2$ nm[26,27].

The presented STM images in Figs. 1f and 2 are measurements of the local density of states (LDOS), which may be considered either in the local state approximation as $\rho_{local}(E, \mathbf{r})$ at the tip position $\mathbf{r}$, for states without long-range translation invariance, $\rho_{local}(E, \mathbf{r}) \propto \Sigma_{i=1,N}|\psi_i(E_i, \mathbf{r})|^2\delta(E - E_i)$; or in the quasiparticle approximation $\rho_{QP}(E, \mathbf{r}) \propto \Sigma_k|\psi_k(\mathbf{r})|^2\delta(E - \varepsilon(\mathbf{k}))$, where $\varepsilon(\mathbf{k})$ is the energy of all the electrons with different wavevector $\mathbf{k}$ that interfere locally at position $\mathbf{r}$. The latter case is relevant when we discuss the electrons in the unconfined 1H layer. It applies also to the interfering itinerant ET-confined electron eigenstates with different $\mathbf{k}_i$ but the same energy $E(\mathbf{k})$. In either case, the resulting

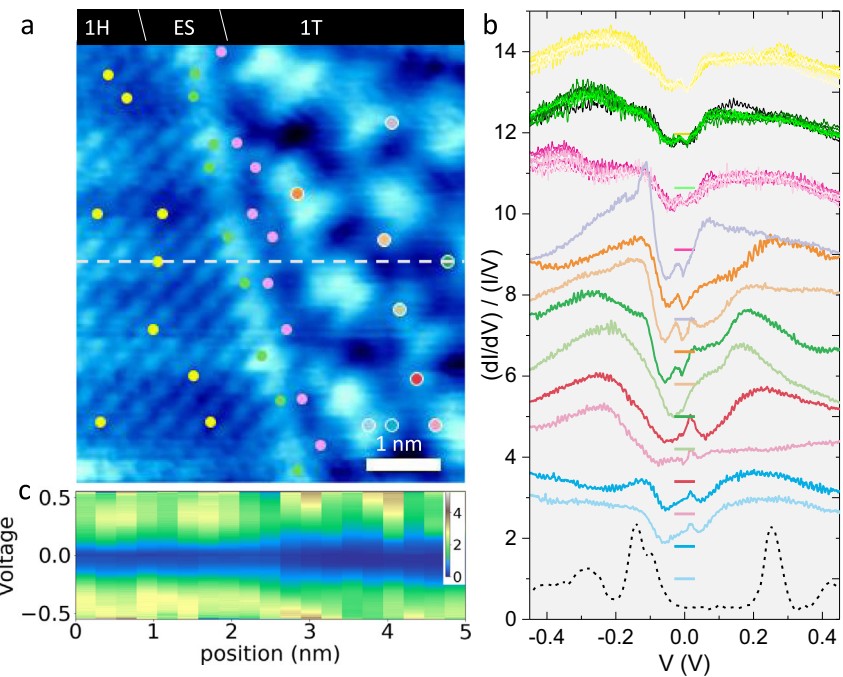

**Fig. 3 NDC curves of the 1H layer outside, the 1T layer inside, and the ES of an ET. a** An STM image at $V = -0.8$ V with indicated positions where STS curves are recorded, using color coded dots. **b** The STS curves (in corresponding color) for the 1H layer, the edge state (ES) inside, and outside the ET boundary, and the 1T phase inside the ET. The zero level is indicated in each case by a line in the same color code. The STS of the CCDW state of a 1T-TaS$_2$ monocrystal at 4 K is shown for comparison (dotted). **c** An STS line scan showing the LDOS (color scale) across the ET boundary. The opening of a pseudogap in the 1H polytype is clearly observed, which increases in the 1T layer inside the ET.

LDOS patterns inside the ET can be observed by STM topography at constant current. The QI is superimposed on the fine sub-nm structure of the atomic orbitals (see, for example, the high-resolution image in the insert to Fig. 1f). This detailed orbital structure of surface atoms is related to structural effects and is not of present concern, so we shall focus primarily on the mesoscopic QI patterns.

Large-area Fourier transforms of the inside and outside of the ETs, shown in Fig. 1g, h respectively, reveal both crystal lattice (CL) peaks and CCDW superlattice (SL) peaks. The areas outside the ETs give sharp FFT peaks corresponding to the 1H CL and weak SL peaks from the 1T-TaS$_2$ CCDW layer underneath (Fig. 1h). Inside the ETs, we see an additional intensity that is almost uniformly distributed between the CCDW FFT peaks. This is attributed to QI features, and the periodic ordering along the inside edges of the ETs (labeled E).

In Fig. 2 we show a representative set of STM images of the inside of ETs, with inner ET dimensions ranging from $l = 8a$ (2.64 nm) to $38a$ (12.35 nm), where $a$ is the CL constant of 1T-TaS$_2$. Universally, we observe that the electrons try to form commensurate order at the center. The CCDW superlattice structure is significantly distorted, however, particularly in small ETs. For the smallest ETs such as the one with $l = 8a$ (Fig. 2a), a single, deformed dot is visible at the center of the ET. Already with $l = 10a$ (Fig. 2b), the pattern with 3 maxima adjusts to the preferred CCDW order of either L or R chirality. In Fig. 2c, d, we clearly see very different QI textures in ETs with the same $l$, which is likely caused by small geometrical imperfections and/or initial conditions that results in different electron trajectories within ETs of equal size. In the same vein, two ETs with $l = 28a$ (Fig. 2e) created simultaneously nearby to each other show quite complicated but closely matching, mirror images of opposite chirality[28]. A remarkable feature of these ETs, particularly well visible in the R structure in Fig. 2e is the nontrivial QI pattern in the corner

which does not fit the CCDW order. As the size of the ETs further increases, the polaron pattern approaches the CCDW super-structure. Thus, for $l = 38a$ the CCDW fills almost the entire ET, with QI distortions visible only at the edges.

Remarkably, the CCDW charge order within the ETs is not in register with respect to the CCDW order of the layer below. (The latter is visible through the surrounding 1H monolayer, and is emphasized by the mesh in Fig. 2h.) Moreover, neighboring ETs also have different register (Fig. 2f). This implies that the effect of the underlying CCDW and lattice potential is not sufficiently strong to force the CCDW order in the top layer. The QI patterns appear to be determined by the ET boundary, not by the inter-layer coupling as has been suggested in bulk crystals[29].

The tunneling spectra inside, outside and across the edge of an ET at 80 K are presented as the normalized differential con-ductance (NDC), (dI/dV)/(I/V) in Fig. 3. The recorded positions are color coded. For comparison, we also show a 1T-TaS$_2$ CCDW bulk crystal spectrum at 4 K showing the characteristic upper and lower Hubbard bands (UHB and LHB) at $-0.15$ and $+0.25$ V, and CCDW-derived bands at $-0.28$ and $+0.42$ V[30].

Overall, the states within $\pm 0.32$ eV of the Fermi level corre-spond to Ta-5$d$ zone-folded sub-bands of the 1T-TaS$_2$ CCDW phase[31], and pristine Ta-5$d$ bands of the 1H monolayer which give rise to the observed NDC. A line scan across the ES (Fig. 3c) shows the spatial variation of LDOS across the ET boundary, which can be compared with the naive band diagram in Fig. 1e. As the STM tip moves across the ES, the screening length 1–2 nm limits the sharpness of the features. Concurrent with the band alignment between 1T and 1H phases, charges will self-organize at the boundary, resulting in the observed ES. On the 1H side, we observe a uniform pseudogap, which we associate with the CDW state nearby (below 75 K).

A most striking feature of the data is the high degree of spatial homogeneity of the NDC curves *outside* the ET (the 1T layer) and

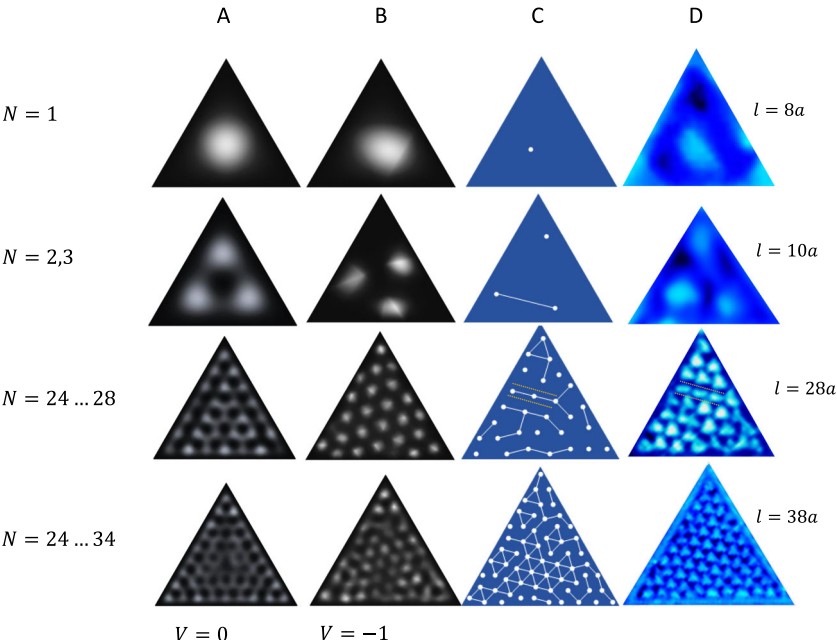

**Fig. 4 A comparison of QB, QB + V and CLG model predictions for integrated LDOS with experiment for ETs of different sizes with different sides *l*.**
**a** Column A represents the QB solution (V = 0) for different summed states (N), corresponding to $V_B \simeq 0.8$ V. **b** Column B shows the QB + V model eigenstates with = −1, and **c** column C is the CLG MC calculation result at 1/13 filling. **d** Column D shows experimental images for *l* = 8 *a*, 10 *a*, 28 *a* and 38 *a*. Note the domain wall predicted for *l* = 28 *a* (col. C), and corresponding STM image (col. D) shown by dashed lines.

of the edge state itself. This is contrasted by the huge spatial spectral variations inside the ET occurring on a scale of 1 nm. Outside the ETs, beside the broad bands, a small, asymmetric gap-like feature is visible around ±20 mV with peaks at ± 50 mV. This feature is similar to the one reported in single layer 1H-TaSe$_2$ on epitaxial bi-layer graphene substrates at 5 K[32] and monolayer 2H-TaS$_2$ on graphene[18], where it is was attributed to the CDW gap of the 1H monolayer. Considering that the data were recorded at 80 K, this would suggest that in the monolayer, the CDW gap is already present above the $T_c = 75$ K of the bulk material. However, no modulation is visible in the FFTs corresponding to the 3 × 3 period of the 1H-TaS$_2$ CCDW (Fig. 1g), implying that there is no long-range CDW order in the 1H layer at 80 K. In some positions we also observe signatures of the characteristic peak at +0.25 V that corresponds to the UHB of bulk 1T-TaS$_2$, which are attributed to the CCDW in the layer below. On the 1T side of the ES, inside the ET, the pseudogap size increases significantly.

Analysing the STS map in more detail at selected points, we note that the ES shows remarkable homogeneity along both sides of the boundary (green and pink dots respectively). The NDCs on the outside edge (the bright border, green) are very similar to the 1H monolayer (yellow). However, the broad peak at −0.21 V (yellow) splits into two, at −0.15 and −0.25 V (green). Inside the ET, the ES peak at −0.25 V shifts further to −0.28 V (pink), but no other significant differences are observed.

Inside the ET, the NDC curves vary significantly from spot to spot. A number of sharp peaks are observed that have no parallel in bulk 1T or 2H-TaS$_2$. Such large spatial variations of LDOS are indeed expected for a confined system with no long-range order. For example, the observation of nodal domains in space is one of the expected features for a QB[33] that modify the CCDW pattern. The sharp features on the energy scale ± 0.1 V are thus attributed to eigenstates of the confined nanostructure.

Investigating the effect of different bias voltages, a set of images at the ET boundary are shown in the Supplementary information. The detailed pattern and contrast change with V,

but the correlation-localized polarons remain in place, in accordance with modeling of the correlated electron state. The contrast changes most when the scanning bias voltage is close to zero, as the contribution of the occupied polaronic states to the LDOS is small and we only see the states very close to the Fermi level.

## Discussion

Within a noninteracting picture, electrons confined within ETs are expected to exhibit canonical QB behavior. On the other hand, the presence of strong correlations localizes electrons in a commensurate structure[12], filling all available space. The noninteracting and strongly interacting pictures are at odds with each other. To investigate the dichotomy, we will compare the observed LDOS patterns with theoretical modeling ranging from noninteracting quantum billiard description to strongly interacting classical gas. We find that the strongly correlated approach is more appropriate but fails to fully describe the QI patterns. Finally, in an attempt to combine the itinerant electron picture with correlations, we calculate electron density patterns that show spatial localization textures using full quantum correlated electron calculation using exact diagonalization methods, finding a remarkable propensity for localization as commensurate filling is approached.

The quantum billiard with inter-layer interactions[12]. For an ideal ET with multiple noninteracting electrons, the levels are filled subject to the Pauli principle. The calculated integrated LDOS based on solutions of the Schrödinger equation are compared with the STM images for different ETs in Fig. 4, column A). The integrated LDOS is given by $\Sigma_{N=n..m}|\psi_N|^2$, where the integers n and m indicate the range of eigenstates for the summation. By choosing appropriate n and m (by inspection), the predicted integrated LDOS patterns show the correct number of maxima within the ET, but the pattern is always symmetric with respect to the ET shape, and parallel with the edges, which the experimental patterns are not. While qualitatively describing

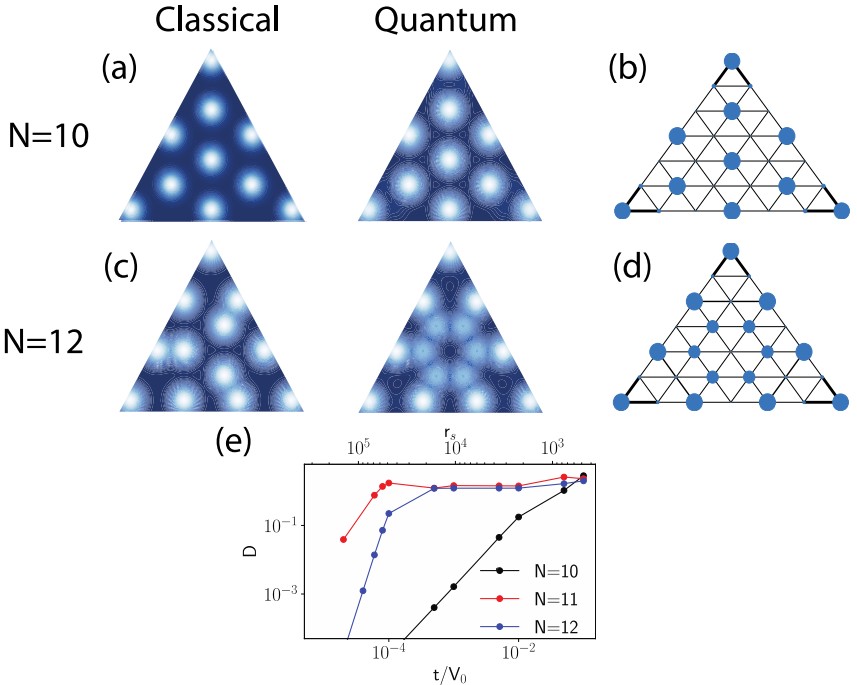

**Fig. 5 Density plot of the charge distribution for spin-less particles on a triangle with $L = 7$.** The first row shows the commensurate case with $N = 10$ electrons. The second row shows the *incommensurate* case with $N = 12$ electrons. The first column represents the classical case with $t/V = 0$. The second column (**a**, **c**) shows the quantum case with $t/V = 0.01$. The size of the dot is a measure of the electron density. The last column (**b**, **d**) represents the quantum $t/V = 0.01$ solution in a graph representation where the thickness of the bond represents a relative contribution to the total kinetic energy and a size of the dot the electronic density. **e** The norm of the density difference distribution D (see the main text) for various number of electrons N versus the ratio between the hopping integral and the interaction $t/V$ or the Wigner-Seitz radius $r_s = 1.92e^2/\hbar v_F$, and $v_F$ is the Fermi velocity.

the integrated LDOS patterns, the QB approach fails to describe the unusual features of the experimentally observed integrated LDOS (column D of Fig. 4).

The energy scale of the states is given by the confinement energy $E \sim \frac{N^2\hbar^2\pi^2}{2m^*l^2}$. Assuming $m^* \sim m_e$, for the smallest ET with side $l = 2.15$ nm $(8a)$, the first four levels $N = 1\dots 4$ have energies ranging from $E \approx 0.08 - 1.26$ eV, while in the largest ET shown here, with $l = 12.8$ nm $(38a)$, for $N = 1\dots 21$, $E \approx 0.0022 - 0.97$ eV. A more realistic effective mass $\sim 3\,m_e$ would compress the energy scale, increasing the number of experimentally observed levels in the STS range $\pm 0.5$ V. The temperature broadening at 80 K is ~7 meV, so the levels are likely too closely spaced to be experimentally resolved. Nevertheless, the energy scale of the observed spectral features (Fig. 3b) is in qualitative agreement with the experiment.

Next, we introduce a periodic potential $V(x, y)$ with magnitude $V_B$ in the QB calculation, $-\frac{\hbar^2}{2m}[\Delta_{x,y} + V(x, y)]\psi(x, y) = E\psi(x, y)$, to account for the periodic lattice distortion created by interaction of the top layer with the CCDW in the layer below. The results are shown in Column B of Fig. 4. Now the 2D pattern corresponding to the CCDW is tilted at 13° with respect to the ET edges, and the crystal axes. As expected, for a significantly large $V(\mathbf{r})$, the calculated integrated LDOS pattern follows the CCDW ordering at the center of the ETs quite well. The periodic potential also introduces a gap in the integrated LDOS which bears resemblance to the observed STS gap (Fig. 3). However, the calculation fails to reproduce the complex features and malleable nature of the integrated LDOS patterns at the edges. It also completely fails to reproduce the observed domain walls. (The predicted integrated LDOS patterns and spectra for the QB, and QB + V models are described in the Supplementary information).

Classical correlated electrons confined within an ET. A correlated electron model is needed to account for the malleable

features in the electronic order within ETs. Within a charge-lattice gas (CLG) Monte-Carlo (MC) calculation, classical point charges, subject to screened Coulomb repulsion can move via thermal hopping on an atomic lattice shaped in the form of an ET (Fig. 4c). A crucial parameter in the presented modeling is the filling, defined as the number of electrons divided by the number of lattice sites $f$. At various magic fillings, such as $f = \frac{1}{13}$ for the case of bulk 1T-TaS₂, the model predicts an electronic superlattice which is perfectly commensurate with the underlying atomic lattice[12]. Confining the system size to a small triangle inevitably introduces edge effects and distortions into the configurational ordering of electrons. This is simply due to the fact that a $\frac{1}{13}$ electronic lattice does not match the edges of the triangle as shown in Fig. 4c, requiring the particles to accommodate. (For more details concerning the model refer to the Supplementary information.) Here we outline the similarities between experimental observations and the simulated electronic configurations (Fig. 4, column C): (1) The model predicts the spontaneous formation of two chiralities of the 1/13 electronic lattice at angles $\pm 13, 9°$ with respect to the ET edges, which are both experimentally observed. In contrast to previous calculations where the external potential fixed the angle of the QI pattern, here it emerges as a nontrivial consequence of many-body correlations. (2) Smaller triangles induce stronger edge effects into the electron configuration, to the point of entirely breaking up the expected 1/13 electronic lattice. Electrons at the triangle's edge align with the edge and since the edges are close one to another, there is no room for a proper 1/13 lattice to emerge. The resulting configuration is still largely influenced by strong correlations, as electrons on average are $\sqrt{13}$ atomic lattice spacings apart. This is shown by the connecting lines in Fig. 4 (column C). (3) Edge effects in larger triangles are diminished towards the center, as shown for $l = 38a$ in Fig. 4 (column D), and the 1/13 Wigner

crystal lattice emerges. However, near the edges the electrons still align with the edge of the ET.

In the simple classical calculation above, all charges are equivalent and should ideally have the same LDOS and thus appear identical under a tunneling microscope. However, the experimental STM images show various irregular, elongated and triangular shapes within the triangles that are sometimes aligned in rows (e.g. Fig. 4) which such modeling cannot describe.

Quantum correlated electrons confined within an ET. For an understanding of the departures from the simple CLG model, we need to consider the itinerant correlated electrons which follow quantum billiard trajectories inside the triangle. To account for the interplay of the strong electronic repulsions and their itinerant nature we perform a quantum many-body calculation on small ET using the exact diagonalization of the spin-less fermions with the long-range interaction:

$$H = -t\sum_{\langle i,j \rangle} c_i^\dagger c_j + \sum_{i \neq j} V(|i-j|)[n_i - \bar{n}][n_j - \bar{n}], \qquad (1)$$

where $c_i$ ($n_i$) is the annihilation (density) operator at site $i$, $t$ is the hopping parameter, ($V(|i-j|)$) is the Yukawa interaction. To ensure charge neutrality, we have subtracted the uniform background charge density n (see Methods for details). While in the classical limit the relevant parameter is filling $N$, the quantum extension introduces another dimensionless parameter $t/V$ that governs the transition between the localized ($t/V = 0$) and the delocalized regime ($t/V \gg 1$). In agreement with the classical simulation, there exist special fillings where the electrons form commensurate fillings of the ET. Due to the size restrictions in the many-body simulation, we consider an ET with the lattice size $L = 8$ and the number of electrons $N = 10$ corresponding to the special filling f = 1/3, see Fig. 5a). In this commensurate situation, the state is extremely robust against the delocalization which can be observed by comparing the density distribution in the quantum $t/V = 0$ and the classical case $t/V = 0.01$, see Fig. 5a). The analogous comparison of the density distribution for the incommensurate filling exhibits a strong redistribution of charges forming nontrivial QPI and their delocalization tendency is driven by lowering the kinetic energy via the closed loops QPI, see Fig. 5c, d.

As a more quantitative measure of the electron delocalization with respect to the classical limit we introduce the parameter $D = \sqrt{\Sigma_i [\langle n_i \rangle - \langle n_i \rangle^{t=0}]^2}$. A comparison for different fillings shows that incommensurate states ($N \neq 10$) are orders of magnitude more susceptible to delocalization than commensurate ones (Fig. 5e). The geometrical constrains of ET can therefore dramatically shift the delocalization transitions and this resolves the apparent contradiction how we can observe delocalized QPI patterns in small structures while the corresponding bulk situation would be localized.

In larger ETs, electron trajectories are modified by correlations within the center, and boundary conditions at the barriers which cannot be understood solely on the basis of semi-classical correlated polaron packing, or free-electron QB. However, correlated electron modeling quite successfully predicts the appearance of domain walls in the CCDW structure forced by the confinement, which cannot be explained by any free-electron QB model. The observation of mirrored, but slightly different QPI patterns within ETs of the same size reveals that the QPI patterns are emergent, self-organized many-body electronic states. The fact that very small changes in geometry and/or barely perceptible imperfections on the atomic level give rise to dramatic changes of the QPI patterns is consistent with chaotic behavior. However, the imperfections in the ET construction (Fig. 1) are expected to change the detailed QI patterns, but not the observed generic features. We showed that for special fillings the system is robust toward these small imperfections, and it

would be interesting to understand the implications of these correlations on the many-body spectrum and the time-evolution in light of recently observed many-body scars in quantum simulators[11,34]. The understanding of behavior and interaction of itinerant electrons with correlation-localized polarons in intertwined textures opens the way to microscopic electronics devices with correlated quantum materials. Moreover, the intertwined orders visible in the ET QPI patterns may help in understanding quantum materials in which a coexistence of itinerant and polaronic correlation-localized carriers is observed, reconciling the seeming dichotomy of different experiments that observe itinerant states (e.g. ARPES, quantum oscillations[35]) and localized states (e.g. optics[36,37], STM[38]) within the same material under seemingly identical conditions.

## Data availability

All of the data supporting the conclusions are available within the article and the Supplementary Information. Additional data are available from the corresponding author upon reasonable request.

## Code availability

The code used in this article is available from the corresponding author upon reasonable request.

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

## Acknowledgements

We wish to acknowledge discussions with Tomaž Prosen, single crystals grown for this work by Petra Sutar, funding from the Slovenian Research Agency (ARRS), projects P1-0040, N1-0092 and young researcher grants, P17589 and P08333. D.G. acknowledges the support by ARRS under Programs No. J1-2455 and P1-0044. The Flatiron Institute is a division of the Simons Foundation. This project has received funding from the European Union's Horizon 2020 research and innovation program under the Marie Skłodowska-Curie grant agreement No 701647.

## Author contributions

J.R., I.V., Y.G., and D.M. conceived the experiments. J.R., Y.V., I.V., P.A., and Y.G. conducted the STM measurements and analyzed the data. J.V., D.G., V.K., and D.M. performed the theoretical calculations. D.M. wrote the paper and supervised the project.

## Competing interests

The authors declare no competing interests.
