## [Peer Review File · Nature Communications]

REVIEWER COMMENTS

Reviewer #1 (Remarks to the Author):

The manuscript by Ravnik and coworkers reports scanning tunneling microscopy data on equilateral “quantum billiards” (or equilateral triangles – ET) defined in the top layer of TaS₂ multilayer system, thanks to a laser-induced local phase change. Such systems, due to their size, are strongly affected by quantum confinement. Furthermore, the strong correlations between electrons in a charge density wave system further complexify the picture. In this framework, the experimental data provided in this work are certainly valuable, and potentially bear signatures of new physics given the peculiarity of the system. This could warrant publication in Nature Communications, if a convincing demonstration is made that the new physics is indeed present, e.g. through modelling/simulation and clear comparison between simulation results and experimental results.

However, as detailed below, I regret to say that I am not convinced at all by the data analysis, and the claims made by the authors about their data. My main criticisms are the following:

1) the authors invoke a lot of different models, each of them with hypothesis that are not fully discussed, and in each case some qualitative connection between the outcome of the simulation and the experimental data. However, I do not see clearly what is better described by a given model or another, when comparing simulation and experimental results, and, in turn, what is the main message that can be concluded from the accumulation of models and simulation results.

2) I do not see in any of the figures comparing data and model the kind of correspondance that one could expect in a system that can be imaged at atomic scale. For example, in Fig. 2f, the authors claim that “the red line emphasize the reginal similarity of the classical trajectory with the QI pattern”. I do not see such a similarity at all.

3) The latter problem also leads to “overclaims”. For example, “quantum scars” are precisely defined objects (see e.g. E. Heller’s works), and to claim that they have been observed requires to verify some criteria (see e.g. in Ref. 9 of the paper, which is, by the way, not an STM study, contrary to what the authors write).

4) All the simulated structures are perfect... what about defects (or inhomogeneous contrast) visible in the vicinity of the ETs, which could be at the origin of asymmetric distributions of STM contrast inside the ETs? defects in the plane of the ETs are visible in Fig. 1f, but in most other pictures, data is

only shown inside the ET, so it's impossible to judge whether the environment of the ET is indeed "perfect". Furthermore, defects could also be present in the underlying layers, and the authors have no information about that. Defects in the alignment of the periodic structure visualized in STM could be present underneath the ET, for example, and lead to contrast "distortion" in the STM images.

Reviewer #2 (Remarks to the Author):

Ravnik et al. report the observation of the quantum interference effects of correlated electrons confined in monolayer 1T-TaS₂ equilateral triangles nanostructures created by femtosecond laser-quenched with scanning tunneling microscopy. This work offers a new playground in the study of the confinement effect on the correlated electrons, even Wigner/Mott crystal ground state. It is worth publishing in Nature Commun. However, the following points should be addressed:

1. The work functions of 1T-TaS₂ and 2H-TaS₂ polytypes used in this paper are from Ref 23, which are the bulk case. The monolayer 1H-TaS₂ on top of bulk 1T-TaS₂ is a different system, and the work function would change. The confinement effect of ET nanostructures and slicing between the ET nanostructure and 1T-TaS₂ underneath, which will also change the work function. With STM, local work function can be detected. The authors should offer more information about the work function, as it is important for the band alignment (Fig 1e) and the explanation of confinement effect of ET nanostructure.

2. Line 18 on Page 6 states that UHB and LHB at +0.15 and -0.22 V, but it is not consistent with the black dash line in Fig 3e.

3. QI features in the manuscript are mainly acquired at 0.8V, which integrates DOS from fermi level to 0.8 eV. Many states far from UHB are included. The author should offer STS instead. And mapping at the energy around UHB and LHB are also important to show how QI features evolve with energy. This is also important to prove the features are from QI, but not just from the distorted lattice of the ET nanostructure.

4. Phase shift of the ET nanostructure was observed, indicating the stacking order between the ET nanostructure and 1T-TaS₂ underneath change. According to L. Ma et al. Nat. Commun. 7, 10956(2016) and T. Ritschel et al. Nat. Phys. 11, 328 (2015), interlayer stacking is a decisive factor in determining the electronic structure of 1T-TaS₂ ground state. Fig3 shows non-zero states appear in

the original bulk 1T-TaS₂ Mott gap, and the STS looks totally different with the black dash one. Will the correlated electrons exist in a lattice distorted, phase shift ET nanostructure?

5. Does the QI feature in ET nanostructure exist below 75K? Will monolayer 1H-TaS₂ go into CCDW phase?

6. In Figs 1g,h, and the caption, $1/13$ SL should be corrected as $1/\sqrt{13}$ SL.

7. For the case of two ET nanostructure sharing a corner point (Fig 1f, Fig 2h), what does the classical trajectory look like? Can the model reproduce QI features in these cases?

8. What is QB short for in the manuscript?

Response to reviewer's comments.

Reviewer #1 (Remarks to the Author):

The manuscript by Ravnik and coworkers reports scanning tunneling microscopy data on equilateral “quantum billiards” (or equilateral triangles – ET) defined in the top layer of TaS₂ multilayer system, thanks to a laser-induced local phase change. Such systems, due to their size, are strongly affected by quantum confinement. Furthermore, the strong correlations between electrons in a charge density wave system further complexify the picture. In this framework, the experimental data provided in this work are certainly valuable, and potentially bear signatures of new physics given the peculiarity of the system. This could warrant publication in Nature Communications, if a convincing demonstration is made that the new physics is indeed present, e.g. through modelling/simulation and clear comparison between simulation results and experimental results.

Authors: We thank the referee for recognizing the new physics and for providing a valuable report. We concur with the main points and have taken care to address the presented issues.

However, as detailed below, I regret to say that I am not convinced at all by the data analysis, and the claims made by the authors about their data. My main criticisms are the following:

1) the authors invoke a lot of different models, each of them with hypothesis that are not fully discussed, and in each case some qualitative connection between the outcome of the simulation and the experimental data. However, I do not see clearly what is better described by a given model or another, when comparing simulation and experimental results, and, in turn, what is the main message that can be concluded from the accumulation of models and simulation results.

Authors: To make our approach clearer, we added a more concise clarification and the purpose for testing each of the models presented, using sub-headings. The underlying hypothesis is presented in each case, and a comparison with experiments, discussing where each model works and where it fails. We made it clear why we decided to use the free electron model as well as the correlated electron model and clarify the main message that the combined model of correlated itinerant electrons is the best approach in the end, albeit limited by current computer capabilities.

2) I do not see in any of the figures comparing data and model the kind of correspondance that one could expect in a system that can be imaged at atomic scale. For example, in Fig. 2f, the authors claim that “the red line emphasize the reginal similarity of the classical trajectory with the QI pattern”. I do not see such a similarity at all.

Authors: We don't show correspondance between calculated scars and experiment, because this was not the aim of the paper (as clearly stated in the introduction). Fig. 4 shows a direct comparison of three different models ranging from free electrons to the other limit of strongly correlated localized states (Columns A-C) compared with experimental data (column D) for different size triangles. The correspondance is discussed, as are the advantages and shortcomings of each model. There is no modelling of scars in the sense of comparing classical and quantum trajectories (see also next point 3), nor are any claims made regarding correspondance of scar models with data.

Careful examination of Fig. 2f shows that the electron density map is clearly correlated with a portion of the classical trajectory indicated by the red line. While the correlation is suggestive, scars are not specifically ‘proven’ (see below), so we concur with the referee that it may be misleading, and omit Fig. 2f.

3) The latter problem also leads to “overclaims”. For example, “quantum scars” are precisely defined objects (see e.g. E. Heller's works), and to claim that they have been observed requires to verify some criteria (see e.g. in Ref. 9 of the paper, which is, by the way, not an STM study, contrary to what the authors write).

Authors: In fact, we never claimed to have proven scars, nor have we aimed at proving them (no specific trajectories or time dynamics has been investigated). Thus, we feel that it is unfair by the referee to imply ‘overclaims’.
Scars are presented in the beginning to introduce the subject of confined electron trajectories. We then immediately state in the introduction: ‘*However, with strongly interacting electrons such patterns are not expected due to their tendency for localization.*’ Scars are not discussed anywhere later in the text, except with reference to further work (Discussion).

We emphasize that scars are based on non-interacting electrons and are too narrow a concept (interpreted strictly along the notions introduced by Heller) to describe our correlated microscopic system, and novel approaches are needed to understand the new physics, which we present. Observing periodicity in the probability of specific many-body states within an ET, or a detailed analysis of the full eigenvalue spectrum in relation to spatial imaging in a large enough ET would prove the existence of correlated-electron features akin to quantum scars, see Ref. 11. However, this is beyond the scope of this paper. In the conclusion, a sentence is added with reference to scars in future work. But we removed the (already very tentative) reference to scars from the abstract to avoid confusion.

The text referring to ref. 9 is corrected by replacing 'tunneling' by 'gate'. Thank you for pointing out the error.

4) All the simulated structures are perfect... what about defects (or inhomogeneous contrast) visible in the vicinity of the ETs, which could be at the origin of asymmetric distributions of STM contrast inside the ETs? defects in the plane of the ETs are visible in Fig. 1f, but in most other pictures, data is only shown inside the ET, so it's impossible to judge whether the environment of the ET is indeed "perfect". Furthermore, defects could also be present in the underlying layers, and the authors have no information about that. Defects in the alignment of the periodic structure visualized in STM could be present underneath the ET, for example, and lead to contrast "distortion" in the STM images.

Authors:

An important argument against an extrinsic source of ordering, such as defects in alignment being responsible for the observed interferences (discussed at the end of p. 5) is shown by the case of $l=27a$, where we state explicitly; 'two ETs shown in Fig. 2fe created simultaneously nearby to each other show quite complicated but closely matching QI patterns, including domain walls in mirrored directions, and overall opposite overall chirality' (pp.5,6). (The raw image which corresponds to Fig 2e that substantiates this claim is included in the revised SI.) This shows that the QI in ETs of the same size is primarily a result of self-organization. The minor differences – which are also apparent – were attributed to imperfections.

This observation also shows unambiguously that the layer below is not causing the observed patterns. We see from the raw image that the layer underneath is all of the same orientation. (This point was already discussed in the original MS with reference to Fig. 2 h where the phase of the electron order in the ET is compared with the layer below). The discussion rules out that the layer underneath, or surrounding layer is causing the observed patterns.

Furthermore, inside the ETs, a $\sqrt{13} \times \sqrt{13}$ commensurate lattice of opposite chirality (this is why we show a few such cases of different size in Fig. 2) cannot be induced by the inter-layer coupling with the commensurate layer below. Even in a perfect world there is a misfit – which is nicely observed in the large triangle in Fig. 2g, and commented upon on p.6, the boundary conditions imposed upon the commensurate correlated state force the ordering at the center of the ET. At the edges QI patterns form, which try to conform to the 13 degree angle mismatch between the correlated $\sqrt{13} \times \sqrt{13}$ structure and the ET sides which are along the crystal axes.

In the theoretical discussion of quasiparticle trajectories, certainly imperfections are a valid concern. However, an important conclusion from the interacting models is that for special fillings relevant in the experiment the strong interactions lead to emergent structures that are very robust against imperfections. We never claim to have a perfect environment, but the salient features of the data are reproduced well, further supporting our claim of new emergent many-body self-organization.

We thus conclude that the observed features are not a result of defects in the alignment or other imperfections surrounding the ETs or the layer below, but are self-organized QI patterns.

Reviewer #2 (Remarks to the Author):

Ravnik et al. report the observation of the quantum interference effects of correlated electrons confined in monolayer 1T-TaS₂ equilateral triangles nanostructures created by femtosecond laser-quenched with scanning tunneling microscopy. This work offers a new playground in the study of the confinement effect on the correlated electrons, even Wigner/Mott crystal ground state. It is worth publishing in Nature Commun.

Authors: we appreciate the comments by the reviewer. We have answered all of the points and made the necessary changes to the manuscript and the SI.

However, the following points should be addressed:

1. The work functions of 1T-TaS₂ and 2H-TaS₂ polytypes used in this paper are from Ref 23, which are the bulk case. The monolayer 1H-TaS₂ on top of bulk 1T-TaS₂ is a different system, and the work function would change. The confinement effect of ET nanostructures and slicing between the ET nanostructure and 1T-TaS₂ underneath, which will also change the work function. With STM, local work function can be detected. The authors should offer more information about the work function, as it is important for the band alignment (Fig 1e) and the explanation of confinement effect of ET nanostructure.

Authors: This is a good point. To answer this question, we include a line scan across the ET boundary, which shows the spatial variation of the local DOS. Considering the known screening length at the boundary between metal and insulating phases of 1T-TaS₂ (refs. 24, 25), which limits the resolution to 1~2 nm, and the discussion in par. 2 p3, and in the SI, the correspondence with the schematic (Fig. 1e) and the measured line scan is clear. In particular, the plot shows the band alignment and reveals the crossover from the 1H-structure surroundings to wider-gapped 1T-structure inside the ET. In response to the referee's comment on band alignment, we now include a discussion of this pseudogap structure in the 1H layer and the self-organization of the charges at the boundary which results in the observed edge state. The line scan complements the STS spectra in showing the DOS structure near E_F. The line scan is added to figure 3 in the revised MS, and an accompanying discussion on p. 7.

2. Line 18 on Page 6 states that UHB and LHB at +0.15 and -0.22 V, but it is not consistent with the black dash line in Fig 3e.

Authors: Thank you for pointing this out. The + and – were reversed in the text by mistake.

3. QI features in the manuscript are mainly acquired at 0.8V, which integrates DOS from fermi level to 0.8 eV. Many states far from UHB are included. The author should offer STS instead. And mapping at the energy around UHB and LHB are also important to show how QI features evolve with energy. This is also important to prove the features are from QI, but not just from the distorted lattice of the ET nanostructure.

Authors: Thank you for this valuable comment. The overall picture indeed changes when changing the bias voltage. However, the changes are mostly in contrast, but not much in the shape of the observed pattern. While we do already show the STS curves in the main text, we agree with the suggestion that showing images taken at different voltages is also important. Thus we have added a section to the supplementary material, showing an edge of an ET at different voltages. One can see a change in detailed pattern and the contrast, but correlation-localized charges remain in place, in accordance with modelling of the correlated electron state. The contrast changes most when the scanning voltage is very close to zero, as the contribution of the occupied polaronic states is very low and we only see the density of states very close to the fermi level. This is the most obvious when scanning the domain walls of the hidden CDW state (images are added to the SI).

In this context we note that lattice distortions in response to local charge ordering are inevitable. The electrons are often considered as polarons, as mentioned in the introduction. But there is no reason or evidence that the lattice would be distorted a priori.

4. Phase shift of the ET nanostructure was observed, indicating the stacking order between the ET nanostructure and 1T-TaS₂ underneath change. According to L. Ma et al. Nat. Commun. 7, 10956(2016) and T. Ritschel et al. Nat. Phys. 11, 328 (2015), interlayer stacking is a decisive factor in determining the electronic structure of 1T-TaS₂ ground state. Fig3 shows non-zero states appear in the original bulk 1T-TaS₂ Mott gap, and the STS looks totally different with the black dash one. Will the correlated electrons exist in a lattice distorted, phase shift ET nanostructure?

Authors: We discussed this point already above, (R1C4), and previously with reference to fig 2e. First of all, the phase of the ordering inside ETs is different in each case, and many different combinations occur. Moreover, two chiralities can often be seen next to each other – (as can be seen in the raw image in the SI), so apparently the layer below does not drive the ordering. We conclude that the layer underneath may have an effect on the detailed QI structure, but is not driving the spatial ordering.

5. Does the QI feature in ET nanostructure exist below 75K? Will monolayer 1H-TaS₂ go into CCDW phase?

Authors: This is a very relevant question and we agree that it is important to discuss also the transition of the H polytype. Since the manuscript is focusing on the electron ordering in the other (1T) polytype, we have decided to not add the discussion to the main text, but rather add a full paragraph with a figure to the supplementary material. In short, the answer to both questions is: yes. The QI features in the ET are present also at temperatures below 75K. At the same time, we can see the 3x3 CDW in the 1H regions outside the triangles, which does not seem to influence the ordering within the triangles.

6. In Figs 1g,h, and the caption, $1/13$ SL should be corrected as $1/\sqrt{13}$ SL.

Authors: Here we meant $1/13$ filling. But it's more common to use $\sqrt{13} \times \sqrt{13}$, so we changed the caption as suggested.

7. For the case of two ET nanostructure sharing a corner point (Fig 1f, Fig 2h), what does the classical trajectory look like? Can the model reproduce QI features in these cases?

Authors: This is a very interesting question and the triangles are indeed not always of perfect shape. In addition to the mentioned point contact between the two triangles, we also observe various other shapes, which can be described by two or more overlapping triangles and even more complex (compound) shapes. We have on purpose limited our discussion only to small simple triangles, where the borders have a significant impact to the ordering and where the shape is easy enough to model. The very simple trajectories may easily become chaotic by changing the shapes or adding imperfections to them. This might be interesting to show, so we have added a section to the supplementary material, showing the experimental images of more complex areas, commenting on their order.

8. What is QB short for in the manuscript?

Authors: We thank the reviewer for pointing this out. A definition of the abbreviation was added.

REVIEWERS' COMMENTS

Reviewer #1 (Remarks to the Author):

I think that the authors took my comments into account and provided convincing answers to my questions. The new version of the manuscript is therefore much better, in my opinion. It can be published in Nature communications, provided that the author address the few minor comments listed hereafter:

- 1) Page 5 – “we observe that the LDOS tries to adapt to the ET boundaries” – I do not really see what this sentence means. Can the author find another formulation?
- 2) Page 5 – “In the same vein, two ETs with $l=27a$ (Fig. 2f) created simultaneously (...)” I guess the author mean “ $l = 28a$ (Fig. 2e)”
- 3) Page 6 – “This does not fit the CCDW order and is strongly suggestive of a quantum interference.”. Invoking “a quantum interference” is abusive here, in my opinion: demonstrating an interference should rely on the observation of an oscillating behaviour which is not the case here.
- 4) Page 6 – “The QI patterns appear to be determined by the ET boundary conditions, not by the inter-layer coupling”. This is again a relatively imprecise statement, as in (1). What do you mean by boundary conditions? is it the confinement barrier height? or the size/geometry of the ET? Or any particular property of the ET boundary itself? (then this should be more clearly stated/discussed).
- 5) Page 7-8 : in Fig. 3 please specify the tip voltage for the STM image.
- 6) Page 12-13 : “The fact that very small changes in geometry and/or barely perceptible imperfections on the atomic level give rise to dramatic changes of the QPI patterns is consistent with chaotic trajectories which originate not only from imperfect ET shapes, but also interaction with crystal lattice fluctuations and inter-particle correlations.” I think this is an overstatement: the authors do not actually discuss these aspects in the paper (imperfections, crystal lattice fluctuations...) so it seems a bit stretched to relate observed changes in QPI patterns to parameters that are not characterized. Can the authors propose another formulation?

Reviewer #2 (Remarks to the Author):

The response has addressed the concerns properly.

Response to comments and suggestions of Referee 1:

1) Page 5 – “we observe that the LDOS tries to adapt to the ET boundaries” – I do not really see what this sentence means. Can the author find another formulation?

Response: The sentence was simplified to read: “Universally, we observe that the electrons try to form commensurate order at the center.”

2) Page 5 – “In the same vein, two ETs with $l=27a$ (Fig. 2f) created simultaneously (...)” I guess the author mean “ $l = 28a$ (Fig. 2e)”

Response: Indeed, this was an error. It is corrected.

3) Page 6 – “This does not fit the CCDW order and is strongly suggestive of a quantum interference.”. Invoking “a quantum interference” is abusive here, in my opinion: demonstrating an interference should rely on the observation of an oscillating behaviour which is not the case here.

Response: The offending formulation was changed by extending the previous sentence and removing the reference to quantum interference: ‘...non-trivial QI pattern in the corner which does not fit the CCDW order.’, thus removing any contentious or misleading statements.

4) Page 6 – “The QI patterns appear to be determined by the ET boundary conditions, not by the inter-layer coupling”. This is again a relatively imprecise statement, as in (1). What do you mean by boundary conditions? is it the confinement barrier height? or the size/geometry of the ET? Or any particular property of the ET boundary itself? (then this should be more clearly stated/discussed).

Response: In fact, we meant to say: “The QI patterns appear to be determined by the ET boundary, not by the inter-layer coupling”. (i.e. not boundary *conditions*). Nothing profound is meant by this statement. It is only a summary of the arguments above pertaining to the importance of inter-layer.

5) Page 7-8 : in Fig. 3 please specify the tip voltage for the STM image.

Response: The value (-0.8V) was inserted in the text.

6) Page 12-13 : “The fact that very small changes in geometry and/or barely perceptible imperfections on the atomic level give rise to dramatic changes of the QPI patterns is consistent with chaotic trajectories which originate not only from imperfect ET shapes, but also interaction with crystal lattice fluctuations and inter-

particle correlations.” I think this is an overstatement: the authors do not actually discuss these aspects in the paper (imperfections, crystal lattice fluctuations...) so it seems a bit stretched to relate observed changes in QPI patterns to parameters that are not characterized. Can the authors propose another formulation?

Response: The statement is made more precise by omitting the second part: “The fact that very small changes in geometry and/or barely perceptible imperfections on the atomic level give rise to dramatic changes of the QPI patterns is consistent with chaotic behaviour.”